# Chromosome-Level Genome Assembly and Circadian Gene Repertoire of the Patagonia Blennie *Eleginops maclovinus*—The Closest Ancestral Proxy of Antarctic Cryonotothenioids

**DOI:** 10.3390/genes14061196

**Published:** 2023-05-30

**Authors:** Chi-Hing Christina Cheng, Angel G. Rivera-Colón, Bushra Fazal Minhas, Loralee Wilson, Niraj Rayamajhi, Luis Vargas-Chacoff, Julian M. Catchen

**Affiliations:** 1Department of Evolution, Ecology and Behavior, University of Illinois, Urbana-Champaign, IL 61801, USA; angelgr2@illinois.edu (A.G.R.-C.); ljwilso2@illinois.edu (L.W.); nirajr2@illinois.edu (N.R.); jcatchen@illinois.edu (J.M.C.); 2Informatics Program, University of Illinois, Urbana-Champaign, IL 61801, USA; bfazal2@illinois.edu; 3Laboratorio de Fisiología de Peces, Instituto de Ciencias Marinas y Limnológicas, Universidad Austral de Chile, Valdivia 5090000, Chile; luis.vargas@uach.cl; 4Centro Fondap de Investigación de Altas Latitudes (IDEAL), Universidad Austral de Chile, Valdivia 5090000, Chile; 5Millennium Institute Biodiversity of Antarctic and Subantarctic Ecosystems (BASE), Universidad Austral de Chile, Valdivia 5090000, Chile

**Keywords:** S. American róbalo, Eleginopidae, basal notothenioid, monotypic, ancestral notothenioid proxy, genome structure, circadian rhythm

## Abstract

The basal South American notothenioid *Eleginops maclovinus* (Patagonia blennie or róbalo) occupies a uniquely important phylogenetic position in Notothenioidei as the singular closest sister species to the Antarctic cryonotothenioid fishes. Its genome and the traits encoded therein would be the nearest representatives of the temperate ancestor from which the Antarctic clade arose, providing an ancestral reference for deducing polar derived changes. In this study, we generated a gene- and chromosome-complete assembly of the *E. maclovinus* genome using long read sequencing and HiC scaffolding. We compared its genome architecture with the more basally divergent *Cottoperca gobio* and the derived genomes of nine cryonotothenioids representing all five Antarctic families. We also reconstructed a notothenioid phylogeny using 2918 proteins of single-copy orthologous genes from these genomes that reaffirmed *E. maclovinus’* phylogenetic position. We additionally curated *E. maclovinus’* repertoire of circadian rhythm genes, ascertained their functionality by transcriptome sequencing, and compared its pattern of gene retention with *C. gobio* and the derived cryonotothenioids. Through reconstructing circadian gene trees, we also assessed the potential role of the retained genes in cryonotothenioids by referencing to the functions of the human orthologs. Our results found *E. maclovinus* to share greater conservation with the Antarctic clade, solidifying its evolutionary status as the direct sister and best suited ancestral proxy of cryonotothenioids. The high-quality genome of *E. maclovinus* will facilitate inquiries into cold derived traits in temperate to polar evolution, and conversely on the paths of readaptation to non-freezing habitats in various secondarily temperate cryonotothenioids through comparative genomic analyses.

## 1. Introduction

The perciform suborder Notothenioidei consists of non-Antarctic and Antarctic lineages, historically classified in eight taxonomic families [1]. Three basal families—Bovichtidae, Pseudaphritidae, and Eleginopidae diverged along the southern coasts of Southern Hemisphere land masses and never experienced polar climate adaptation. The other five families—Nototheniidae, Artedidraconidae, Harpagiferidae, Bathydraconidae, and Channichthyidae—evolved and diversified in situ in the oceanographically and thermally isolated Southern Ocean (SO). They formed an adaptive radiation and became the dominant fish group endemic to the freezing SO [2,3]. As a rare marine species flock [4,5] that successfully overcame and flourished in otherwise uninhabitable icy freezing conditions for ectothermic teleosts, the Antarctic clade of notothenioid fishes (cryonotothenioids) have been the subject of decades of numerous studies covering all aspects of its biology and evolution.

The extraordinary ecological success of cryonotothenioids resulted from the evolutionary innovation of macromolecular antifreeze glycoproteins (AFGP) [6,7], which avert death from freezing by preventing internal ice nucleation [8,9] along with biochemical and physiological adaptive changes that enable cellular and life processes at rate-depressive and protein denaturing subzero temperatures [10,11,12]. Becoming highly adapted and specialized to chronic cold apparently came at the cost of developing extreme stenothermy, with extant cryonotothenioids exhibiting greatly reduced thermal tolerance [13,14]. In parallel, certain essential traits have been lost, hypothesized to result from relaxation of selective pressure in the polar environment for their maintenance. The most ostensible are the loss of the classic inducible heat shock response (HSR) across cryonotothenioids [15,16,17] presumably due to the lack of large thermal variations in the cold-stable SO, and the loss of hemoglobin and red blood cells in the icefish family Channichthyidae [18,19], where stably cold and oxygen-rich SO waters presumably provide a reliable oxygen source that made simple diffusion a viable mechanism for oxygen transport.

Understanding the cold adaptive and specialized states in cryonotothenioids continues to be an active pursuit, particularly in light of the vulnerability of these stenothermal fishes to increasing threats of climate change. Thus, added to the historical interest of how cryonotothenioids evolved cold adaptation and specialization to dominate the SO fish fauna, are pressing questions regarding whether their derived polar character harbors sufficient plasticity and adaptive potential to survive the changing contemporary SO environment. To this end, the fast trajectory of generating high quality genome assemblies of species across Notothenioidei promises to be an accelerator in our ability to address a wide range of important questions, including variations in genetic diversity, mutation rates, transposable element mobilization [12,20,21], genome structures [22], protein genes experiencing positive or relaxed selective pressures [23], and historical and extant demography (by whole genome resequencing of populations) shaped by Antarctic glacial cycles [24], among others, that would inform on the degree of resilience of the cryonotothenioids under climate change scenarios. Seventeen draft genomes of notothenioids with various degrees of completeness have been published since 2014. A large advancement in genome resources in form of an additional two dozen new draft genomes are becoming publicly available [21].

Investigations of derived polar characters in cryonotothenioids by necessity would require having an appropriately related non-Antarctic species representing the ancestral state for comparison, to discern and/or test hypotheses of Antarctic-specific trait changes. Species of the three basally divergent non-Antarctic notothenioid families—Bovichtidae, Pseudaphritidae, and Eleginopidae—all constitute candidates as ancestral proxy. A chromosome level genome of one bovichtid species *C. gobio* has been published [25], but it represents the most basally divergent lineage, which may be less informative in deciphering the temperate to polar transition. The Patagonia blennie (locally known as róbalo), *E. maclovinus*, the monotypic species of the family Eleginopidae, long recognized as the taxonomic and phylogenetic direct sister species of the Antarctic clade [26,27,28] would serve as the most fitting closest ancestral state for evaluating trait changes and reconfigurations shaped by Southern Ocean selective pressures during cryonotothenioid evolution. A few studies have utilized *E. maclovinus* as the ancestral reference, namely the nature of the native chaperome [29], loss of the classical heat shock response [15], proliferation of transposable elements and a number of developmental traits during cold adaptation [20], and the state of the AFGP orthologous genomic region prior to the emergence of the AFGP genotype [22]. Apart from these more visible traits, there are expectedly myriad subtler trait changes across the genetic blueprint, requiring a well assembled and gene-complete genome to decode.

A draft genome of the Patagonia blennie has previously been reported using Illumina short reads, and is thus lacking contiguity [20]. In this study, we generated a highly contiguous and gene-complete, chromosome-level genome assembly of *E. maclovinus* using PacBio CLR (continuous long read) sequencing and HiC scaffolding. We found substantial differences in the genome structures between *C. gobio* and *E. maclovinus*, while the *E. maclovinus* genome and those of nine cryonotothenioids species representing all five families of the Antarctic clade share a larger degree of structural conservation. We further evaluated *E. maclovinus* as an ancestral reference for addressing functional trait change in cryonotothenioids by characterizing the network of genes that choreograph the circadian rhythm, a trait that is likely distinct due to disparate light regimes between temperate and high polar latitudes. We found *E. maclovinus* to share a more similar pattern of retention of functional circadian genes with cryonotothenioids than *C. gobio*. These genome scale findings solidify *E. maclovinus* as an evolutionarily strategic sister species of the Antarctic clade, and its chromosome-level genome an important resource, in furthering our inquiries into cryonotothenioid evolution, past and present.

## 2. Materials and Methods

### 2.1. Specimen Collection and Sample Preservation

Specimens of *E. maclovinus* were collected from two Chilean coastal water locations. In June 2008, several young individuals (~80–100 gm) were collected using rod and reel from the Patagonian waters near Puerto Natales, Chile (51.82° S, 72.63° S). We are deeply thankful to Mathias Hüne for the collection. Blood cell DNA of one male individual from this collection was used for whole genome sequencing in this study, and liver from the same fish was collected for Hi-C sequencing. Fresh whole blood was obtained from the live specimen on ice through the caudal vein using needle and syringe. The blood cells were spun down, washed with notothenioid fish ringer (0.1 M sodium phosphate buffer, pH8.0, adjusted to 420 mOsm with NaCl) to remove plasma proteins, then embedded in 1% low-melt agarose blocks (Bio-Rad Plug Kit #1703591) (Bio-Rad, Hercules, CA, USA) to preserve DNA integrity. The embedded blood cells were lysed exhaustively in situ using 1% lithium dodecyl sulfate in notothenioid ringer, then stored in 0.5M EDTA, pH 8.0, at 4 °C until use. Post bleeding, the fish was immediately dissected on ice to obtain the liver and other tissues. The excised liver and tissues were cut into small cubes, and each tissue was submerged in 20 mL glass vials filled with −20 °C pre-chilled 90% ethanol (made with pure 100% ethanol and ultrapure water). The ethanol was replaced with a fresh volume twice within the next 2–3 h, followed by storing the tissue vials in a non-frost free −20 °C freezer. The handling and sampling of fish complied with Protocol #07053 approved by the University of Illinois Institutional Animal Care and Use Committee (IACUC).

The other collection, a cohort of about 30 small juvenile (~10–20 gm) *E. maclovinus* specimens were collected for a prior study [15] using a small seine in the near-shore water of Valdivia, Chile during the austral winter of 2014. Specimens were used for the study at the Laboratorio Costero de Recursos Acuáticos Calfuco of the Universidad Austral de Chile. Collection and aquarium maintenance of specimens were conducted with approved protocol 11/10 by the Comité de Bioética Animal, Universidad Austral de Chile. The post-experiment specimens were quickly dissected on ice to obtain the liver and other internal organs. These small tissues were submerged in 20 mL glass vials filled with −20 °C pre-chilled 90% ethanol. Each dissected fish carcass was then submerged in a 50 mL tube of −20 °C pre-chilled 90% ethanol. Handling and sampling of fish were conducted in compliance with Protocol #12123 approved by the University of Illinois IACUC. All samples from both collections were air freighted back to the University of Illinois on dry ice, then stored in a non-frost free −20 °C freezer until use. In this study, pectoral fin tissue from twelve of the ethanol preserved carcasses of juvenile *E. maclovinus* from the second collection were taken for RNA isolation to obtain full-length transcripts using PacBio (Pacific Biosciences, Menlo Park, CA) Iso-Seq sequencing.

### 2.2. HMW DNA Preparation and CLR Sequencing

To isolate agarose-embedded high molecular weight (HMW) DNA, each agarose block was melted (70 °C, 10 min), and digested with 2 units of β-agarase (New England Biolab, Ipswich, MA, USA) (42 °C, one h). The released DNA was extracted using the Nanobind CBB Big DNA Kit and protocol (Circulomics, Baltimore, MD, USA; acquired by Pacific Bioscience in 2021) with adjustments in reagent volumes. DNA concentration was determined with Qubit Broad Range DNA (Invitrogen, Carlsbad, CA, USA) fluorometry, purity with Nanodrop One (Thermo Fisher Scientific, Waltham, MA, USA) and MW and integrity with pulsed field electrophoresis (CHEF Mapper, Bio-Rad, Hercules, CA, USA). The recovered *E. maclovinus* DNA showed high quality preservation (>50 to >150 kbp) through 12 years of storage prior to sequencing.

The PacBio CLR (Continuous Long Read) library was prepared using unsheared HMW DNA, size selected for ≥45 kbp inserts using the BluePippin (Sage Science, Beverly, MA, USA) and sequenced on two SMRT cells on Sequel II for 30 h of data collection at the University of Oregon Genomics & Cell Characterization Core Facility (GC3F). A total of 10.12 million reads were obtained, with an average length of 10.86 Kbp, and a read N50 of 22.07 Kbp. Total read length was 110.7 Gb, equivalent to 142× genome coverage, based on a genome size of ~780 Mb previously estimated with flow cytometry of erythrocytes (mean c-value 0.798 pg) [20]. Based on assembled length of 606 Mbp (see Section 3), the genome coverage was 180×.

### 2.3. Hi-C Library and Sequencing

For genome scaffolding, a chromatin proximity ligation Hi-C library was prepared from the ethanol preserved liver of the same male *E. maclovinus* specimen by Phase Genomics Inc. using its Proximo Hi-C kit. The library was sequenced for paired-end 150 bp on Illumina NovaSeq6000 (Illumina, San Diego, CA, USA) at the Roy J. Carver Biotechnology Center, University of Illinois, Urbana-Champaign, which generated 204.5 million paired-end reads.

### 2.4. Genome Assembly

Following a published assembly optimization strategy [30], we generated subsamples of the raw PacBio CLR read data covering various distributions of read lengths and depths of coverage, and assembled each subsampled dataset to arrive at the most optimal contig-level assembly. These contig-level assemblies were generated using *WTDBG2* v2.5 [31]. The assemblies were then self-corrected using the *GenomicConsensus*
*arrow* v2.3.3 tool from PacBio (https://github.com/PacificBiosciences/GenomicConsensus, accessed on 24 October 2019), which identifies and fixes small indels and other errors in the assembled genome based on the sequence consensus present in the aligned raw reads. After polishing, each separate assembly was assessed based on contiguity metrics using *QUAST* v4.4 [32] and gene completeness metrics using *BUSCO* v3.0.1 [33] with the actinopterygii_odb9 reference ortholog set. The best contig-level assembly, i.e., the most contiguous and gene-complete, was obtained from reads 10–40 Kb in length, and a depth of coverage of 64×.

The optimal contig-level assembly was then integrated with the Hi-C data to generate chromosome-level scaffolds. The *3D-DNA* program from *Juicer* v1.6.2 [34] was used to identify Hi-C junctions and perform integration and ordering of the contigs into super-scaffolds. This chromosome-level assembly was reassessed for contiguity using *QUAST* and for gene-completeness using *BUSCO* v5.1.3 [35] with the actinopterygii_odb10 reference ortholog set. Additional manual inspection and curation of this scaffolding process was conducted using a conserved synteny analysis (methods described below), e.g., verifying that structural variants were not limited to the boundaries of a contig/scaffold, or that the large-scale organization of the chromosomes was supported by patterns of within-species synteny. Any manual changes made to the assembly were propagated to the corresponding structure (AGP), annotation (GTF), and sequence files (FASTA) using a custom Python program, as described in [22].

### 2.5. Repeat and Protein-Coding Gene Annotation

Repeat elements in the genome were annotated by first building a de novo repeat library with the *RepeatModeler* v2.0.2a pipeline [36], using the *BuildDatabase* option and the NCBI database as input (-engine ncbi). We then used the *RepeatModeler* option to generate the final repeat library, using *ltrharvest* [37] to perform the discovery of LTRs (-LTRStruct). We then extracted teleost-specific repeats using the famdb.py script from *RepeatMasker* (-species Teleostei) and combined these teleost-specific repeats with the de novo repeat library generated by *RepeatModeler*. Final annotation and masking were completed using *RepeatMasker* v4.1.2-p1, using our custom library (-lib).

To annotate protein coding genes, we used the RNAseq data available in NCBI SRA under accession number SRX2523921 we previously generated to build a de novo reference transcriptome for *E. maclovinus* [15]. The RNAseq reads were aligned to the reference assembly using *STAR* [38] v2.7.1.a (--runMode alignReads). These alignments were used as input to the *BRAKER* v2.1.6 annotation pipeline [39,40] in transcript mode (--bam). In addition, we also ran *BRAKER* in protein mode, using the *orthodb* v10.1 [41] zebrafish reference protein sequences as input (--prot_seq). The output of both *BRAKER* runs were then aggregated using *TSEBRA* v1.0.1 [42], obtaining a curated set of protein-coding gene annotations supported by both transcript and protein evidence.

### 2.6. Conserved Synteny Analysis

We used the *Synolog* software [43,44] to evaluate conserved synteny and determine orthologous chromosomes between *E. maclovinus* and other notothenioid species. A total of ten notothenioid genomes were used for comparisons against *E. maclovinus*, which include the basal, non-Antarctic *C. gobio* (Bovichtidae) [25], five red-blooded cryonotothenioids across four different families (*Trematomus bernacchii* (Nototheniidae) [21], *Dissostichus mawsoni* (Nototheniidae) [45], *Gymnodraco acuticeps* (Bathydraconidae) [21], *Harpagifer antarcticus* (Harpagiferidae) [21], and *Pogonophryne albipinna* (Artedidraconidae) (NCBI GCA_028583405.1), and four white-blooded icefishes (Channichthyidae; *Champsocephalus esox* and *Champsocephalus gunnari* [22], *Chaenocephalus aceratus* (NCBI GCA_023974075.1), and *Pseudochaenichthys georgianus* [21]). To run *Synolog*, we first reciprocally matched the annotated gene models for all species using the *blastp* algorithm in *BLAST+* v2.4.0 [46]. The resulting *BLAST* outputs and the annotation coordinates then serve as input to *Synolog*, which finds orthologous genes using reciprocal *BLAST* best hits and identifies blocks of conserved synteny based on the coordinates of the matched orthologs along a sliding window. The matching of orthologs is further refined by sequential rounds of synteny block identification, which allows for a more accurate identification of orthologous genes between species in the presence of paralogs.

In addition to these ten notothenioid genomes, we also included comparisons against three teleost outgroup species, including zebrafish (*Danio rerio*), threespine stickleback (*Gasterosteus aculeatus*) and platyfish (*Xiphophorous maculatus*), in order to validate the annotation of genes of interest (genes participating in circadian rhythm, discussed below). In addition, this conserved synteny approach was used to name the *E. maclovinus* chromosomal scaffolds. The *E. maclovinus* chromosomes were numbered according to the orthologous platyfish chromosomes (Appendix A), as this outgroup species serves as an example of the ancestral teleost karyotype of 24 chromosomes [47]. Accession information for the genome assemblies used for the comparative synteny analysis and circadian gene curation (next section) is given in Appendix A.

### 2.7. Notothenioid Species Phylogeny Reconstruction

We leveraged our new *E. maclovinus* genome data and reconstructed an abbreviated species phylogeny of the 11 notothenioids and the two teleost outgroups in this study using genome wide single-copy ortholog genes to reaffirm *E. maclovinus*’ phylogenetic relationship to the cryonotothenioids. We first clustered the annotated protein sequences of the 13 species into orthogroups using the *orthofinder* v2.5.4 package [48,49]. We ran *orthofinder* (-M msa) to generate multiple sequence alignments (MSA) using *MAFFT* v7.310 [50], filtering columns in the alignment composed of >90% gaps and retaining alignments with lengths of ≥500 sites (default settings), followed by tree inference using *RAxML* v8.2.11 (-T raxml) [51]. A total of 2918 single-copy orthologs across the 13 species were identified, which *orthofinder* internally uses to generate a species tree using *STAG* [52] and rooting it using *STRIDE* [53].

The concatenated MSA for these 2918 orthologs were used as input to generate a maximum likelihood phylogeny using *IQ-TREE* v 2.2.2.5 [54]. We ran the iqtree2 executable, selecting the best partition and substitution models for this multi-gene amino acid alignment (--seqtype AA) using -m MFP+MERGE. To reduce analysis runtime, we limited the search to nuclear substation models (--msub nuclear) and testing only the top 10% of partition schemes (--rclusterf 10). Support of the inferred trees was evaluated with 1000 ultra-fast bootstrap replicates (-B 1000) [55]. The resulting tree was visualized, and midpoint rooted using *FIGTREE* v1.4.4 (http://tree.bio.ed.ac.uk/software/figtree/, accessed on 15 May 2023).

### 2.8. Curation of Predicted Circadian Network Genes from Genome Assemblies

We performed a comparative analysis on the presence and absence of 33 genes of known circadian function (Appendix A) in *E. maclovinus* and the 10 abovementioned notothenioid genome assemblies. First, we identified the respective IDs and sequences of these genes in the zebrafish (*D. rerio*); 2 of the 33 are absent in the zebrafish annotation, thus references from the threespine stickleback (*G. aculeatus*) were used. For each query assembly, we located the orthologs of the 33 model teleost reference genes by comparative synteny analysis in *Synolog*. *Synolog* identifies orthologous sequences first by identifying reciprocal *BLAST* best hits between a query and subject species (e.g., *E. maclovinus* vs. *D. rerio*), which are then refined by sequential rounds of conserved synteny identification. A reference gene was denoted as absent from a query assembly when no orthologous sequence could be identified by *Synolog*. Once orthologs were identified in each query assembly, we used a custom Python script (see Data Availability Statement) to extract the corresponding peptide sequences and annotation coordinates for use in downstream analyses.

### 2.9. Phylogenetic Verification of Circadian Gene Orthologs

The query assemblies used for the curation of circadian network genes were not all RefSeq assemblies (see Appendix A) and thus may contain annotation errors. We evaluated the accuracy of a subset of the curated circadian genes—the *arntl/BMAL, Clock, Cry*, and *Per* paralogs, which participate in the core circadian feedback loop using phylogenetic analyses. A MSA for the protein sequences of each gene family and the orthologous sequences from human (Appendix A) and zebrafish were generated with *MUSCLE* [56], and maximum likelihood phylogeny was constructed using *IQ-TREE* v 2.2.2.5 [54]. We ran the iqtree2 executable with --m MPF implementing the model selection tool *modelfinder* [57] to identify the best-fit substitution model for each MSA (--seqtype AA). *modelfinder* identified the best-fit substitution models as follows: *arntl/BMAL*- JTT+F+R3; *Clock*- JTT+F+I+G4; *Cry*- JTT+R4, *Per*- JTT+F+I+G4. We tested the support of each inferred tree using 1000 ultra-fast bootstrap replicates (-B 1000) [55]. Trees were visualized and midpoint rooted using the *USER TREE* menu in *MEGA* v11.0.13 [58].

### 2.10. Fin RNA Isolation, Iso-Seq Transcriptome Sequencing and Curation of Circadian Network Gene Transcripts

Teleost fins are known sites of peripheral circadian clock [59,60]. We evaluated functionality of the predicted circadian network genes in *E. maclovinus* by PacBio Iso-Seq sequencing of fin transcriptome. About 5–10 mg of pectoral fin tissue from each of the 12 ethanol preserved juveniles mentioned above were homogenized in Trizol (Invitrogen, Carlsbad, CA, USA) with 0.5 mm zirconium oxide beads using a Bullet Blender^R^ (Next Advance Inc. Troy, NY. USA), followed by extraction of total RNA according to the Invitrogen protocol. RNA purity and concentrations were initially assessed with the Epoch Take 3 microplate spectrophotometer (BioTek Instruments, Winoosk, VT, USA), and RNA integrity by visualizing ~1 μg of RNA electrophoresed on a 1% formaldehyde agarose gel. Equal amounts (2 µg) of RNA from each sample were pooled into a single sample and treated with DNaseI with added RNase Inhibitor (New England Biolabs, Ipswich, MA), then purified using the Monarch^R^ RNA Cleanup Kit (New England Biolabs). The quality of the cleaned, pooled RNA sample was checked using Qubit Broad Range RNA fluorometry (Invitrogen, Carlsbad, CA, USA) for concentration, Nanodrop One (Thermo Fisher Scientific, Waltham, MA, USA) for purity, and Agilent 5200 Fragment Analyzer (AATI) (Agilent, Santa Clara, CA, USA) for RNA Quality Number; the pooled RNA sample achieved a RQN over 8.

Iso-Seq library construction and sequencing were conducted at the Roy J. Carver Biotechnology Center, University of Illinois, Urbana-Champaign. About 300 ng of the pooled RNA sample was converted to cDNA with PacBio Iso-Seq^R^ Express Oligo Kit (Pacific Biosciences, Menlo Park, CA, USA), and the sequencing library was constructed with the PacBio SMRTBell Express Template Prep kit 3.0. The library was sequenced on one SMRT cell 8M on a PacBio Sequel IIe using the CCS (circular consensus sequencing) mode for 30 h of data collection. Over 3.3 million HiFi reads totaling 6.18 million bases were obtained. Removal of primers, trimming of polyA tails and other clean ups of the HiFi reads, followed by de novo clustering and consensus calling of transcripts were performed using PacBio *SMRTLink* v.11 software, producing a final set of 167,428 high-quality (99.99% accuracy) consensus transcript isoforms.

The coding sequences of the 33 predicted circadian network genes were used as queries to identify their expressed transcripts from the high-quality Iso-Seq transcript dataset by blastn search. Since almost all circadian network genes occur in two or more paralogs, and blastn search would match a given query to all its paralogs, the assignment of the transcript/s specific to each paralog was based on >99–100% nucleotide identity to the query gene model. The transcripts were extracted and aligned with the query gene model to ascertain absence of mutations.

## 3. Results and Discussion

*A priori*, the long recognized position of *E. maclovinus* in notothenioid taxonomy [26] and molecular phylogeny [27,28] as the closest non-Antarctic sister species to the Antarctic notothenioid radiation indicates its suitability as proxy for the ancestral notothenioid state, useful for deciphering evolutionary changes in the Antarctic clade. Our study provides robust whole genome evidence that supports this hypothesis, through generating a long-read, chromosome-level assembly of *E. maclovinus* genome, comparing its genome architecture with the genomes of the basal *C. gobio* and nine derived notothenioids representing all five Antarctic families, a species phylogeny constructed with 2918 genome-wide proteins of single-copy orthologous genes, and comparing its repertoire of circadian network genes against the same ten genomes.

### 3.1. Genome Assembly and Annotation

We generated a chromosome-level genome assembly of the Patagonian blennie using PacBio CLR sequencing and Hi-C scaffolding. The final assembly was 606 Mb in length, smaller than the 744 Mb scaffold-level *E. maclovinus* assembly first described by [20]. The longer length of the original, short read-based assembly could be attributed to its low contiguity (scaffold N50 of 695 Kb), likely reflecting unassembled repetitive regions and alternative haplotypes. Our newly generated PacBio assembly is highly contiguous, with a contig and scaffold N50 of 7.6 Mb and 26.7 Mb, respectively (Table 1). A total of 99.98% of the total sequence length was assembled into 24 chromosome-level scaffolds, in agreement with the 2n = 48 karyotype previously described for the species [61]. In addition to its contiguity, the *E. maclovinus* assembly is 96.5% gene complete, according to BUSCO v5.3.1 analysis. The combination of high contiguity and gene-completeness indicates the smaller size of the PacBio long-read assembly compared to the short-read *E. maclovinus* assembly is not likely caused by portions of the true genome missing. Instead, the updated PacBio assembly is a better representative of the genome architecture of *E. maclovinus*.

A total of 25 thousand protein-coding genes were annotated using a combined proteome- and transcriptome-based annotation. This number is comparable with the 20–30 thousand protein-coding genes described in other notothenioid assemblies [21,22,62], and the 23 thousand genes annotated for the Illumina short-read assembly [20]. With regard to repeat content, this assembly is composed of 33.2% repetitive sequences, characterized from a de novo repeat library. Nearly half of the repetitive fraction (15.8% of the total genome) was annotated as DNA mobile elements, with the second largest classified fraction (5.1% of the genome) being composed of LINE sequences (Table 1). The 33% repetitive fraction of the genome in *E. maclovinus* is smaller than observed in other cryonotothenioids [21,22,62]. In the context of genome evolution across Notothenioidei, this result reinforces the idea that expansion of repeat elements and associated increase in genome size observed in cryonotothenioids [63] is specific to the Antarctic clade. As the direct extant outgroup of the Antarctic clade, the *E. maclovinus* genome likely reflects the ancestral state of the genome architecture in this group.

### 3.2. Conserved Synteny

Using a comparative synteny analysis, we identified conservation in the large-scale genome organization between basal, non-Antarctic notothenioids and highly derived members of the Antarctic clade. The genomes of both *C. gobio* and *E. maclovinus*, both basal, non-Antarctic species, show conservation in size (~600 Mb) and one-to-one correspondence of 24 orthologous chromosomes (Figure 1A). A similar relationship is observed when comparing these two basal species to representative members of the Antarctic clade. When compared to *E. maclovinus*, both the red-blooded *D. mawsoni* (Figure 1A) and the white-blooded *C. gunnari* (Figure 1B) also exhibit clear correspondence across 24 orthologous chromosomes; however, both species exhibit larger genomes sizes, which are linked to the expansion in repeat elements observed in the Antarctic clade [25,63]. This same relationship, one-to-one correspondence between 24 chromosomes and an increase in genome size in cryonotothenioids, is also observed when we compared the *E. maclovinus* genome against other Antarctic species with chromosome-level assemblies, namely *P. georgianus* and *C. esox*. While changes to the number of chromosomes have previously been observed, including independent evolution of sex chromosomes [64,65,66] and extensive chromosomal fusions leading to karyotype reduction [67], this one-to-one relationship is largely seen across both basal and derived notothenioid species, and provides further evidence for an ancestral karyotype of 24 chromosomes in the group.

Focusing on individual chromosomes, we find evidence for numerous changes in the chromosomal structure between basal and derived notothenioids. Several chromosomal rearrangements, including inversions and translocations, are observed between orthologous chromosomes in *C. gobio* and *E. maclovinus* (Figure 1C,D). In accordance with their well-recognized phylogenetic placement—in which *E. maclov*inus is the immediate sister taxon to cryonotothenioids [3,28]—we observe a higher degree of conservation in the chromosome structure and organization between *E. maclovinus* and *C. gunnari*, in which synteny is completely conserved along whole chromosomes (Figure 1C). However, other orthologous chromosomes show extensive evidence of chromosomal rearrangements between these linages (Figure 1D). This finding indicates that, while genome structure may be largely conserved, there is evidence for the independent evolution of chromosomal organization between basal and Antarctic notothenioids. Nonetheless, given the substantial chromosomal differences observed between *E. maclovinus* and *C. gobio*, and the relatively large degree of conservation between *E. maclovinus* and cryonotothenioid genomes, the *E. maclovinus* assembly may be better suited as an outgroup for future genome-wide comparisons in Antarctic notothenioids.

### 3.3. Phylogenomic Analyeses Affirm E. maclovinus as Immediate Sister to Cryonotothenioids

We utilized our new *E. maclovinus* genome assembly, and the other 10 notothenioid genomes and two non-notothenioid reference teleost genomes included in this study to generate a large, genome-wide dataset of 2918 single-copy protein orthologs discoverable from these genomes to reconstruct an abbreviated notothenioid species phylogeny. As expected, the phylogenetic relationships among non-Antarctic and Antarctic notothenioids are in accordance with prior phylogenetic hypotheses of the group [3,21,28]. Members of the Antarctic clade form a single monophyletic group comprising the five cryonotothenioid families, with the Patagonia blennie *E. maclovinus* as the immediate sister taxon (Figure 2), supported with 100% bootstrap proportions. This independent affirmation of the monotypic *E. maclovinus* as the immediate sister taxon to the cryonotothenioids further supports the prevailing acceptance of its suitability as an ancestor proxy when investigating derived adaptive phenotypes in members of the Antarctic notothenioid clade.

### 3.4. Circadian Gene Repertoire in E. maclovinus and Ten Notothenioids

In addition to establishing *E. maclovinus* as a better ancestral reference for whole genome architecture evolution and re-affirming its immediate sister relationship to the cryonotothenioids through phylogenetic reconstruction using genome scale protein data, we further evaluate its suitability as a reference for addressing functional trait evolution in the Antarctic clade. We analyzed the network of genes that choreograph the circadian rhythm, arguably the most crucial endogenous process guiding the tempo of cellular functioning, systems physiology and whole animal activity throughout life.

*E. maclovinus* and other basal non-Antarctic notothenioids are expected to have a usual circadian rhythm based on their evolutionary distribution in temperate latitudes where diurnal and seasonal light-dark cycles predictably occur. The evolutionary status of the circadian process in Antarctic notothenioids, however, is at present completely unknown. The variable photoperiod regimes in the polar region means light-dark zeitgebers, which are major cues that entrain the oscillation of the circadian network, would be irregular for the endemic species. Thus far, circadian rhythms have been investigated in depth for only one Southern Ocean organism, the Antarctic krill *Euphausia superba* [68,69]. In free-running conditions, the krill exhibited endogenous rhythmic transcriptional expression of several circadian clock genes, but at shortened periodicities from the typical 24 h to ~12–15 h. That correlated with similarly shortened periodicities of functional outputs in enzymatic and metabolic activities and diel movements in the water column. These observations were hypothesized to represent a circadian adaptation to the extreme range of polar photoperiodic variability [68,69]. Antarctic krill occupies or is most abundant in relatively lower Southern Ocean latitudes (~53–65° S) [70]. The variability of the polar light regime would reach its utmost for the benthic notothenioid fishes localized in the highest latitude Antarctic coastal waters and embayments such as McMurdo Sound (78° S), where prolonged thick sea surface fast ice cover further reduces, or completely excludes light penetration into their habitats seasonally, leading to drastic reduction or protracted absence of light cues. In the marine realm, water temperature changes and tidal cycles could also time the circadian clock [71]. High-latitude Antarctic coastal sites, however, are nearly always at freezing, showing negligible summer warming [72,73], and circum-Antarctic coastlines are microtidal because their narrow continental shelves with deep shelf slopes do not significantly amplify tidal currents [74]. How the circadian network is entrained in the highest latitude endemic fish fauna and other marine organisms that experience these unique environmental conditions is particularly rich ground for inquiry. For notothenioids it begins with documenting and comparing the repertoire of the circadian network genes between the cryonotothenioids and the non-Antarctic ancestral proxy *E. maclovinus*.

Leveraging publicly available chromosomal or near-chromosomal scale genomes of 10 notothenioids, and the *E. maclovinus* genome in this study, and using a set of 33 circadian network genes from the reference teleosts *D. rerio* and *G. aculeatus*, we curated the orthologs from these 11 notothenioid species that represent seven of the eight families of the Notothenioidei suborder (Appendix A). Figure 3B summarizes the curated circadian gene orthologs found in the notothenioids, and contrasts gene presence and absence relative to the teleost reference set, and between the basal species (*C. gobio* and *E. maclovinus*) and the nine derived cryonotothenioids. The 33 reference circadian genes/paralogs were selected based on their known function in the widely studied circadian system in Drosophila [75] and in mammals [76,77], and that orthologs are present in teleost fish. Figure 3A provides a simplified depiction of the autonomous cellular transcription-translation negative feedback regulatory loops underlying circadian rhythmicity, and the positions of the 33 genes/paralogs in the circadian process. The core loop comprises the phasic interactions of the activator proteins of *arntl/BMAL* and *clock* with the suppressor proteins of *cry* and *per* paralogs (gene names highlighted in light yellow in Figure 3B). Secondary loops involve *CK1e/CK1d*, and the nuclear receptors *nr1d* and *ror* paralogs (gene names highlighted in light green in Figure 3B), whose proteins regulate the core clock genes. *Timeless* protein (not shown in Figure 3A) also participates in a secondary loop, interacting with *per* proteins and others to suppress arntl/BMAL activation of *per*.

Relative to the teleost reference set, several paralogs of the core loop clock genes, namely *arntl1b, cry3b, cry4, and per1a*, as well as *nr1d1* and *rorab* of the secondary loops were absent across all basal and derived notothenioids (Figure 3B), suggesting these six genes were absent in the common ancestor of the suborder. Within notothenioids, the two basal lineages differ in their circadian gene repertoire. The maintained set of genes (27) in *E. maclovinus* is similar to the teleost reference, while *C. gobio* has apparently undergone lineage-specific loss of additional *cry* (*cry1a, cry3a*) and *per* (*per2a, per3*) paralogs since the two basal taxa diverged. Between *E. maclovinus* and the derived cryonotothenioid species, the core clock genes *cry1b, cry2* have undergone Antarctic-specific loss. *Per2a* and *per3* were also lost in the cryonotothenioids, separately from their loss in *C. gobio,* since they are present in *E. maclovinus* the closest sister taxon to the Antarctic clade.

The remaining core clock genes are largely conserved between *E. maclovinus* and the derived cryonotothenioids. However, the *cryptochrome* paralog *cry3a* was not detected by *Synolog* in *D. mawsoni* (nototheniid) or the *P. albipinna* (artedidraconid) genome, suggesting varied losses in the Antarctic clade, or inaccuracy in the current state of the genome annotations of these two species. By manual *BLAST* searches, we recovered a full-length copy of *cry3a* from both species (Appendix A). Through manual annotation, we found *D. mawsoni cry3a* (in HiC_scaffold_10; Appendix A) to be completely conserved in gene structure but contains a 1-nt indel in each of two exons, which would result in a frameshifted protein gene model with little sequence similarity to intact cry3a, thus eluding detection by *Synolog. Cry3a* in the artedidraconid *P. albipinna* (in scaffold CM053253.1; Appendix A) however, is completely mutation free, and thus a structurally intact gene (Appendix A). It has eluded detection likely because the current annotation has split up the gene into two halves, each with a hypothetical protein as annotation. With the manual recovery of *D. mawsoni* and *P. albipinna cry3a*, it appears likely that this core clock gene paralog is preserved across the Antarctic clade. In contrast, manual search for *cry3a* in *C. gobio* draft genome only found an orphan exon 1 in an unplaced scaffold, verifying definitive loss.

Presence of the rhythmic genes of the secondary regulatory loops—*timeless*, and paralogs of *CK1d/e* (casein kinase 1) and *ror* (retinoic-acid-related orphan receptors) is conserved throughout notothenioids (Figure 3B). In contrast, the two paralogs of *nr1d2* (nuclear receptor subfamily 1 group D) appear to have undergone varied loss in different Antarctic lineages. This *nr1d* evolutionary pattern is difficult to interpret, and how much bioinformatics inaccuracy plays a part remains to be evaluated.

### 3.5. Phylogenetic Analyses of Core Regulatory Loop Circadian Genes

Given the annotation pitfalls encountered, we performed phylogenetic analyses to validate the automated annotation of the paralogs of the core clock gene families *arntl/BMAL*, *Clock*, *Cry*, and *Per* from the notothenioids. Besides the teleost reference, human orthologs of each gene family are also included, allowing for assessment of which teleost paralog is the evolutionary functional counterpart. The Maximum Likelihood tree of each circadian gene family is shown in Figure 4. The ML trees recovered each paralog in each of the four circadian gene families as a monophyletic clade with robust (97% to 100%) node support, indicating overall accuracy of the gene curation using *Synolog*. The topology of each gene clade also reflects biologically sensible phylogenetic positions of the human and teleost (*D. rerio*) references as outgroups of the notothenioid clades. Within the notothenioid ingroups, the gene tree topology in most cases matches known species/family phylogeny, with genes of the non-Antarctic *E. maclovinus* and *C. gobio* being the basal branches relative to the Antarctic cryonotothenioids.

However, annotation inaccuracies are also evident in various forms. Most visible is the substantially shorter protein lengths (amino acid numbers indicated with an asterisk, Figure 4) for some species compared to the typical lengths across species of the gene clade. This led to atypical long branches, for example, in the case of several *H. antarcticus* genes including *arntl1a* and *BMAL2* (Figure 4A), and *clock b* and *npas2* (Figure 4B), which also resulted in the harpagiferid oddly nesting within the icefish lineages (Figure 4B). *C. gobio arntl1a* (Figure 4A) is another instance of the error affecting *H. antarcticus clock b* and *npas2*, despite the *C. gobio* genome assembly is a RefSeq genome.

In manual assessments, atypical lengths of the predicted protein, which often also contained divergent sequence segments, were found to likely result from splice junction errors, leading to various structural outcomes. These include reading frame changes and premature truncation of the encoded protein, inclusion of sequences from intron locations, and/or exclusion of *bona fide* coding exons. As an example, *arntl1a* of *P. albipinna*, annotated in the database as hypothetical protein (GeneID JOQ06_002371, Appendix A), was excessively long in length (806 aa) compared to lengths of 620–650 aa of orthologs from other species (Figure 4A), but it was also prematurely truncated. We manually found the C-terminus coding exons in the adjacent unannotated region. The long protein length resulted from the presence of a repeated sequence segment covering a second copy of exon2 through exon9 after a gap in the middle of the annotated gene (Appendix A). This repeated segment was likely due to assembly error, and the repeated exons were annotated as part of the predicted protein, leading to the atypical long length of 806aa. We resurrected the complete and correct gene sequence, which is 638 amino acids long, and which shares high sequence identity with *E. maclovinus arntl1a* and included it in the input dataset for the ML tree (Figure 4A). This is a second *P. albipinna* circadian gene (the first one being *cry3a* described above) with a similar annotation error of splitting the gene. It underscores the state of the quality and annotation inaccuracy of the current public versions of some of the assemblies used in this study, perhaps due to the lack of, or inadequate species-specific supporting transcript evidence. Manual scrutiny to curate the correct coding sequences from these assemblies (even for ReSeq genomes) is essential prior to utilizing them in experimental studies or phylogenetic inferences as was also cautioned by others [80].

The phylogenetic analyses included human circadian genes as references, as their gene functions are well studied through the use of mammalian models, so we may glean insights into the evolution of the circadian network genes and function in the notothenioids. For the *arntl/BMAL* family, members of which form the central circadian pacemaker with a *clock* gene, the notothenioids have three paralogs—*arntl1a*, *arntl2*, and *BMAL2* (Figure 4A) that can participate in this function. The single arntl1 gene—*arntl1a*, which forms a monophyletic clade with human *BMAL1*, would be the putative functional counterpart of the latter, while *arntl2* and/or *BMAL2* would be that of the single human *BMAL2.* For clock genes, humans have two paralogs—*CLOCK* and *NPAS2* (Figure 4B). Notothenioids, like zebrafish, have two duplicated *clocks*—*clocka* and *clockb*, either of which could fulfill the role of the human *CLOCK*. The neurally expressed *npas2* (neuronal PAS domain Protein 2) is similarly a single copy between human and fish. For the cryptochromes, members of which complex with a period protein to regulate the pace setting activity of heterodimerized arntl/BAML and clock proteins, humans have two copies—*CRY1* and *CRY2* (Figure 4C). In contrast, the *Cry* family has diversified substantially in various teleost lineages, with zebrafish possessing eight paralogs (Figure 3B). *E. maclovinus* has five of these paralogs, which were reduced to four in the cryonotothenioids (Figure 3B). *Cry1a*, which is the single *cry1* retained across the cryonotothenioid families, form a monophyletic clade with human *CRY1* (Figure 4C), suggesting it is the functional equivalent for the latter in these Antarctic fishes. *Cry3a*, the immediate sister clade to *Cry1,* and which is also maintained across the cryonotothenioids, potentially can fulfill the same role. It is also sister to *Cry2* in the *E. maclovinus* and *C. gobio,* and thus may also be able to serve the role of human *CRY2.* For period genes, cryonotothenioids have lost *Per3* present in *E. maclovinus* and zebrafish, retaining only two paralogs—*Per1b* and *Per2* (Figure 3B and Figure 4D) which are likely functional counterparts of human *PER1* and *PER2* respectively based on their close phylogenetic relationships (Figure 4D). These inferences will certainly require experimental testing for biological function in future studies, and the analyses here serve as a starting platform for experimental design.

### 3.6. Transcriptional Expression of E. maclovinus Circadian Network Genes

Curation of circadian network genes from genomes serves to catalog the set of potentially active contributors to circadian rhythm generation. To ascertain the functionality of these predicted genes requires transcriptional evidence at minimal. Having transcriptional expression evidence is also crucial as a starting point for testing actual biological function as mentioned above. We thus obtained high-quality long read transcripts for *E. maclovinus* by PacBio Iso-Seq sequencing of fin tissue, where peripheral clock is known to operate in teleost fishes [59,60]. Where Iso-Seq transcript is missing for a given gene, we searched the multi-tissue reference transcriptome assembled from published Illumina RNAseq short reads [15].

In total, we found mRNA expressions for 24 of the 27 predicted circadian genes/paralogs in the *E. maclovinus* genome (Figure 3B; sequences in Appendix A). Of the 24 expressed genes, 20 are supported by full-length cDNAs (complete coding sequences) in the transcriptomes. They are indicated with uppercase letter T (Figure 3B) as follows: 18 (white letter T) are full-length Iso-Seq transcripts, one (red letter T) full-length is from overlapping partial Iso-Seq transcripts and partial transcripts from the RNAseq reference transcriptome, and one (orange letter T) full-length is from overlapping partial transcripts from the RNAseq reference transcriptome. The other four (indicated with white lowercase t) are three *per* paralogs (*per2, per2a, per3*) and one *ror* paralog (*rorb*), with expression evidence as partial transcripts in the RNAseq reference transcriptome.

Transcripts of the remaining three *E. maclovinus* clock genes—*BMAL2, clocka* and *npas2*—are lacking in both the Iso-Seq and RNAseq transcriptome. Their absence can either be due to the genes being inactive, or their mRNA not being represented in the RNA sample that was sequenced. *Npas2* (neuronal PAS domain Protein 2) is the *clock* paralog with demonstrated expression in teleost brain [81,82]. Whether it is expressed in a peripheral tissue such as the fin is unknown, thus its absence in the fin Iso-Seq transcriptome may be due to tissue specificity of expression. The RNAseq reference transcriptome did include whole brain in the pooled RNA sample [15], but its small fractional input (1 of 11 tissues) might not have yielded sufficient sequence coverage to encompass the full complement of circadian network genes. An additional constraint in capturing the full circadian transcriptome is the interlocking phasic up- and down-regulation of gene expression in the different regulatory loops, such that genes that are transcriptionally activated at the time of tissue sampling are more likely to be captured, while those that are repressed or undergoing degradation are not. The tissues used for the two transcriptomes came from the same individuals sampled over an afternoon for the purpose of the published study [15]. To definitively ascertain transcriptional functionality of *BMAL2*, *clocka* and *npas2* will require using RNA from tissues (both fin and brain) sampled at regular intervals over a complete (24 h) circadian time period for additional sequencing.

In sum, the *E. maclovinus* genome contains orthologs of 27 of the 33 reference circadian network genes/paralogs of model teleosts, with gene functionality of 24 validated by transcript evidence predominantly in full length Iso-Seq cDNAs. The five missing circadian genes are also absent in the more basally divergent bovichtid *C. gobio*, reflecting their loss in the common ancestor of Notothenioidei. *C. gobio* further lost *cry1a* and *cry3a* and would be reliant on *cry1b* and *cry2* as the participating *cryptochrome* paralogs in the core regulatory loop. In contrast, functional *cry1a* and *cry3a* are maintained in *E. maclovinus* and gene models are present across species of the Antarctic clade. In the latter, *cry1a* and *cry3a* would likely become the core loop repressors, as *cry1b* and *cry2* were lost after their divergence from *E. maclovinus.* Overall, *E. maclovinus* and cryonotothenioids share more of the same retained circadian genes than between the latter with *C. gobio*. This supports *E. maclovinus* as the better candidate as an ancestral reference for investigating circadian and other functional trait changes during polar evolution of cryonotothenioids. It can also serve as a reference for inquiries into potential changes in the re-colonization of non-freezing environments by secondarily temperate notothenioids.

## 4. Conclusions

The Patagonia blennie *E. maclovinus* occupies a strategically important phylogenetic position in Notothenioidei, as the monotypic species of the basal, non-Antarctic family Eleginopidae and immediate sister lineage to the Antarctic clade of cryonotothenioids. Using PacBio CLR sequencing and HiC scaffolding, our study provided a highly contiguous and gene-complete chromosomal-level assembly for this species. Through comparative whole genome architecture analyses with the genomes of 10 other notothenioids collectively covering seven of the eight notothenioid families, substantial structural divergence can be seen between the two basal species *C. gobio* and *E. maclovinus*, while there is a larger degree of conservation between *E. maclovinus* and derived cryonotothenioids. Our reconstruction of an abbreviated notothenioid phylogeny covering the same species using the largest genome wide dataset to date—2918 proteins from predicted single-copy ortholog genes—solidly affirms *E. maclovinus* as the direct sister species to Antarctic cryonotothenioids. We curated and compared the circadian network genes from *E. maclovinus* and the 10 notothenioid genomes to evaluate the changes in the gene complement during evolution under extreme polar photoperiodic variability and inferred functional roles of the retained circadian genotypes referenced to known functions of human orthologs as informed through phylogenetic analyses. In our new *E. maclovinus* assembly, we documented a repertoire of 27 circadian network genes, and provided expression evidence supporting active transcription for most of them. Comparison of the pattern of circadian gene/paralog retention and loss among the same 10 species used for genome structure analysis, again illustrates greater similarity between *E. maclovinus* and the cryonotothenioids than between *C. gobio* and the latter. Results in this study collectively reinforce unequivocally the suitability of *E. maclovinus* as an ancestral proxy in investigating genome character and functional trait evolution in the derived Antarctic clade during its evolution in extreme polar conditions. We additionally suggest that *E. maclovinus* can also serve as a comparison in understanding the nature of the evolutionary path and mechanisms underlying secondary adaptation to non-freezing environments of various cryonotothenioid lineages that have undergone polar to temperate transition.

## Figures and Tables

**Figure 1 genes-14-01196-f001:**
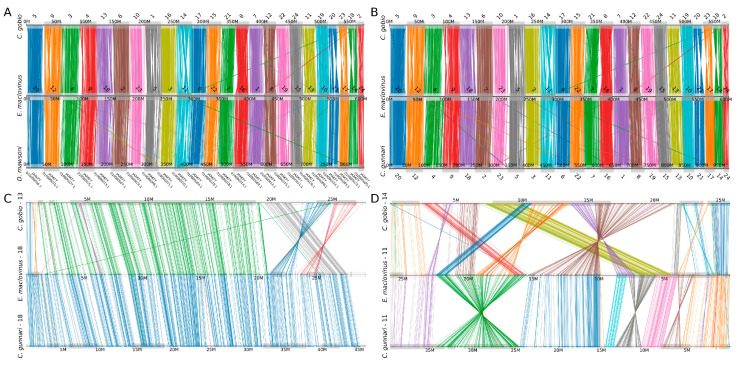
Patterns of conserved synteny across basal temperate and derived Antarctic notothenioid genomes. (**A**) Genome-wide synteny shows a one-to-one correspondence between the 24 chromosomes of the basal temperate *C. gobio* (top), the basal temperate *E. maclovinus* (middle), and the Antarctic red-blooded *D. mawsoni* (bottom). Each line represents an orthologous gene between the compared genomes, color-coded according to their chromosome of origin. (**B**) Genome-wide conserved synteny shows a one-to-one correspondence between the genomes of *C. gobio* (top), *E. maclovinus* (middle), and the Antarctic, white-blooded *C. gunnari* (bottom). Color annotation as described in panel A. (**C**) Conserved synteny between orthologous chromosomes in *C. gobio*-13 (top), *E. maclovinus*-18 (middle), and *C. gunnari*-18 (bottom). Lines represent orthologous genes between the species, color coded according to their respective synteny clusters. Chromosomes are not drawn to scale. Several chromosomal inversions and translocations are observed between *C. gobio* and *E. maclovinus*, while the chromosome organization is fully observed between *E. maclovinus* and *C. gunnari*. (**D**) Conserved synteny between orthologous chromosomes in *C. gobio*-14 (top), *E. maclovinus* -11 (middle), and *C. gunnari*-11 (bottom). Color notation as described in panel C. Several examples of chromosomal rearrangements are observed between the three species.

**Figure 2 genes-14-01196-f002:**
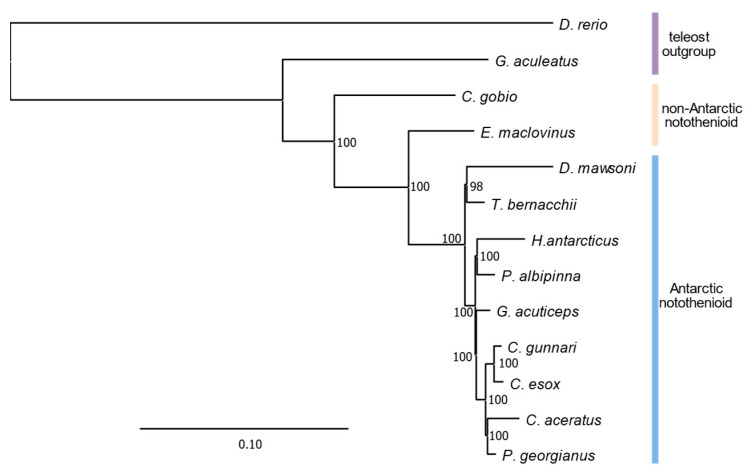
Phylogenetic relationship among non-Antarctic and Antarctic notothenioids. Midpoint-rooted maximum likelihood tree for 11 notothenioids and two teleost outgroup species generated from the protein alignment of 2918 single-copy orthologous genes. Branch lengths describe the number of substitutions per amino-acid site. Ultra-fast bootstrap support values are shown for each non-outgroup node. Vertical colored bars on the right show the placement of Antarctic notothenioids (blue), non-Antarctic notothenioids (beige), and teleost outgroups (violet). The placement of *E. maclovinus* as immediate sister taxon to the Antarctic notothenioid clade is recovered with 100% support.

**Figure 3 genes-14-01196-f003:**
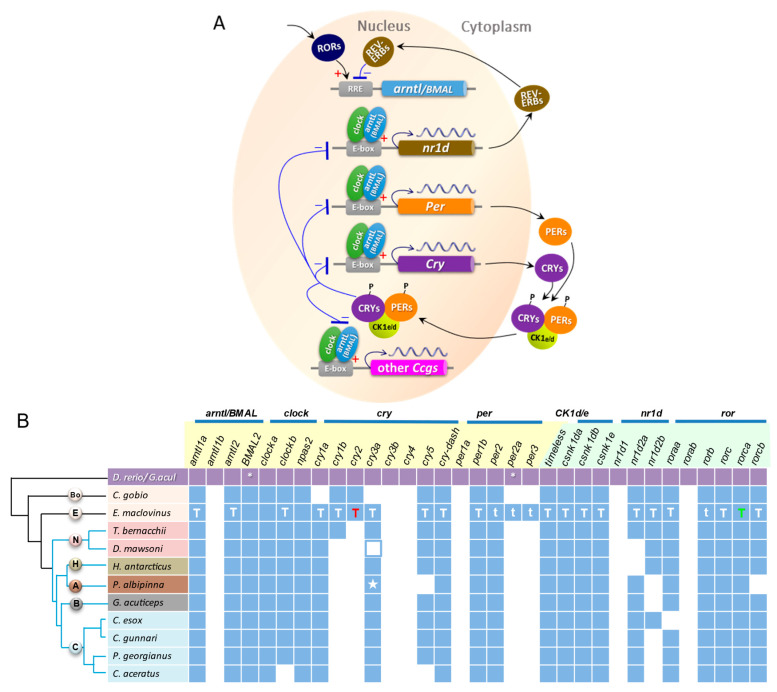
Schematic of circadian regulatory loops and the circadian network genes identified from the genome and transcriptomes of *E. maclovinus* and from genomes across Notothenioidei. (**A**). A simplified depiction of the circadian regulatory loops based on mammalian studies [78,79] relating the positions of the orthologous genes identified from notothenioids in the circadian network. Briefly, the activator core clock proteins arntl/BMAL (aryl hydrocarbon receptor nuclear translocator-like and brain and muscle arnt-like are alternate names) and clock (circadian locomotor output cycles protein kaput) heterodimerize and bind to regulatory elements containing E-box sequences of a large number of clock-controlled genes (*Ccgs*) including the repressor genes *cry* and *per*. The cry and per proteins accumulate in the cytoplasm, interact with each other and the serine/threonine casein kinases *CK1e* and *CK1d*. The complex translocates back into the nucleus and suppresses target gene activation by arntl/BMAL and clock, leading to their own transcriptional repression. *Timeless* protein also interacts with *per* proteins and others to suppress arntl/BMAL activation of per1. The regulation of *arntl/BMAL* and *clock* transcription is in turn effected through a loop involving nuclear receptor proteins REV-ERB and ROR. Activation of *nr1d* by *arntl/BMAL* and *clock*, and of *ROR* through an intervening activator, produces REV-ERB (encoded by *nr1d* paralogs) and ROR proteins, respectively. These compete for their common binding sequence element RORE in *arntl/BMAL* promotor, where ROR initiates and REV-ERB inhibits *arntl/BMAL* transcription. (**B**). Evolutionary status of circadian network genes in genomes of notothenioid fishes identified using 33 reference genes from zebrafish *D. rerio* and three-spined stickleback *G. aculeatus.* All reference genes (violet squares) except two are from *D. rerio*, with * indicating *G. aculeatus* as the source since it is absent in *D. rerio*. Names of the core loop circadian genes are highlighted in light yellow, and the secondary loop rhythmic genes in light green, with paralogs of each gene family grouped under a blue bar. Light blue squares indicate presence of notothenioid orthologs in the genome annotation of each species determined with *Synolog*, and uncolored (white) squares indicate gene absence. A cladogram on the left depicts the phylogenetic relationship of the included notothenioid species, which represent seven of the eight families—Bo (Bovichtidae), E (Eleginopidae), N (Nototheniidae), H (Harpagiferidae), A (Artedidraconidae), B (Bathydraconidae), and C (Channichthyidae). *Synolog* did not detect *cry3* in *D. mawsoni* or *P. albipinna.* Its presence in these two species was determined by manual BLAST search and annotation, which found *D. mawsoni cry3* (blue outlined white square) gene structure to be conserved except for two 1-nt frameshift mutations, and *P. albipinna cry3* (blue square with white star) to be complete and mutation-free. Circadian gene repertoire in the two basal non-Antarctic notothenioids differ, with *E. maclovinus* retaining more of the reference teleost set, while apparent *C. gobio*-specific loss of two *cry* (*cry1a, cry3a)* and two *per (per2a, per3*) paralogs had occurred since the two species diverged. Transcript evidence supporting functionality of the circadian network genes in *E. maclovinus* indicated by the letter “T” or “t” are as follows: T (white)—Iso-Seq transcript, complete cds; T (red)—complete cds from overlapping partial Iso-Seq + partial Trinity assembled RNAseq transcripts; T (green)—overlapping Trinity transcripts, complete cds; t (white)—partial Trinity assembled RNAseq transcript.

**Figure 4 genes-14-01196-f004:**
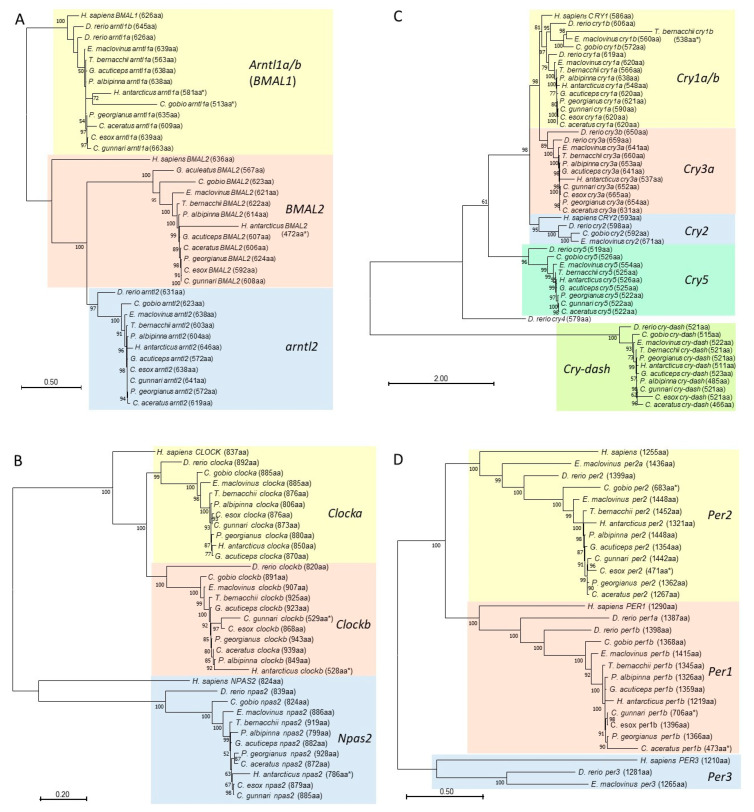
Maximum Likelihood trees for verification of circadian gene orthologs. Maximum Likelihood trees were constructed with *IQ-TREE* v 2.2.2.5 using aligned peptide sequences of annotated circadian network genes from the notothenioid species representing seven families and the teleost reference in Figure 3B. Human orthologs were also included as reference. Gene members of the (**A**) *Arntl/BMAL*, (**B**) *Clock*, (**C**) *Cry*, and (**D**) *Per* gene families, which set the pace of the core circadian feedback loop, were analyzed. By convention, human clock gene names are capitalized. The paralog of each gene family from the sampled species formed a monophyletic clade, highlighted in color blocks and labeled with the gene family name. All four gene trees were midpoint rooted. For the *arntl/BMAL*, the tree was additionally rooted on the *Homo sapiens BMAL1*/teleost *arntl1a* clade to capture the relationships more accurately between members of this gene family. Ultra-fast Bootstrap support values > 50% are shown at the nodes. Amino acid lengths of the predicated proteins are given in parentheses next to species names. Asterisk (*) denotes reduced protein length and/or potential errors in annotation of the predicted amino acid sequences. One species, *D. mawsoni* (family Nototheniidae) was excluded from the analysis due to annotation inaccuracies in almost all of the core clock genes, such that manual re-annotation would be prohibitively laborious. The Nototheniidae family is represented by *T. bernacchii*.

**Table 1 genes-14-01196-t001:** Assembly and annotation metrics for the *E. maclovinus* genome.

Assembly Contiguity		
Scale	Contig	Scaffold
Total Size (bp)	606,099,673	606,289,673
Number of Fragments	406	26
Largest Fragment (bp)	19,993,184	37,057,500
N50 (bp)	7,582,354	26,674,500
L50	27	11
Number of Chromosome-Scale Scaffolds	--	24
Total Bases in Chromosomes (bp)	--	606,192,173
Percent of Assembly in Chromosomes	--	99.98%
* **BUSCO ** * **v5.3.1 Gene Completeness**		
Complete	3513 (96.5%)	
Complete and Single Copy	3476 (95.5%)	
Complete and Duplicated	37 (1.0%)	
Fragmented	12 (0.3%)	
Missing	115 (3.2%)	
Total	3640	
Reference Gene Set	actinopterygii_odb10	
**Gene Annotation**		
Number of Protein Coding Genes	25,081	
**Repeat Annotation**		
Total Repeats	33.18%	
LTR	2.92%	
SINE	0.64%	
LINE	5.06%	
DNA	15.84	
Unclassified	5.72%	

## Data Availability

The raw PacBio and Hi-C Illumina reads used for the *E. maclovinus* genome assembly are available on NCBI under BioProject PRJNA917608. A copy of the *E. maclovinus* reference assembly and annotation, alongside the gene models and annotation for all studied notothenioid species, and copies of all bioinformatic scripts used for analysis are available on DRYAD (https://doi.org/10.5061/dryad.qbzkh18nt).

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
