# Peer review of "Chromosome-Level Genome Assembly and Circadian Gene Repertoire of the Patagonia Blennie Eleginops maclovinus—The Closest Ancestral Proxy of Antarctic Cryonotothenioids"

_genes, 2023, doi:10.3390/genes14061196_

Round 1

Reviewer 1 Report

Editor overridden review feedback: A really nice piece of work on the study of the genome (at the chromosome level) of Eleginops maclovinus which I really enjoyed reading.

I have a few comments and suggestions, which are really aimed at making the manuscript clearer (section 3.3 relative to the circadian gene repertoire in E. maclovinus).

The authors documented a repertoire of 27 circadian network genes in the E. maclovinus genome and provided expression analyses thus  supporting the active transcription for most of them.

Unfortunately as correctly mentioned by authors the evolutionary status of the circadian rhythm in Antarctic notothenioids, is completely unknown for comparison.

However there are some (few) studies on Antarctic organisms which can be briefly mentioned to describe how the Antarctic environment characterised by extreme seasonal changes may influence the expression of circadian genes.

I think some mention of these observations (although speculative) present in the literature should be added in, as it will not take much work to do so and will produce a more comprehensive overview.

This may also help you contextualise your nice results and opens up the possibility of looking at the regulators of circadian rhythms also in Antarctic fish. This is just a suggestion.

Author Response

We agree with the reviewer's suggestion.  We have added a paragraph on the circadian rhythm studies of the Antarctic krill Euphasusia superba, which as far as we know, is the most in depth for Southern Ocean marine organisms.  We additionally highlighted the lack of temperature and tidal variations in the Southern Ocean to cue circadian oscillation, particularly extreme for the highest latitude benthic organisms, such as those in McMurdo Sound (78S),  to set the context for studies of those species that evolved under these unique conditions. The relevant text is in the paragraph in lines 439-467.

Reviewer 2 Report

In my opinion, this work is a very important contribution to understanding the evolution of Antarctic fish. 

Abstract:

-I suggest not including abbreviations in the abstract, such as S. American.

Methods

-In section 2.1 Include in the sampling section how liver was sampled and conserved up to their use for DNA extraction and Hi-Ci library preparation and sequencing

Results and Discussion

-In several lines across section 3.1 and 3.2 E. maclovinus and C. gobio are written without italics.

-In order to support the conclusions and go deeper in the evolutionary discussion, I suggest including a phylogenomic approach using a subset of genes of the BUSCO catalog of orthologs for a general tree, and another one more specific for the conserved circadian network genes.

Author Response

We thank the reviewer's input. Comments are entered here with our response immediately below.

Abstract:

-I suggest not including abbreviations in the abstract, such as S. American.

Response:  It is now spelled out as “South American”

Methods

-In section 2.1 Include in the sampling section how liver was sampled and conserved up to their use for DNA extraction and Hi-Ci library preparation and sequencing

Response: Details have been added in Section 2.1

Results and Discussion

-In several lines across section 3.1 and 3.2 E. maclovinus and C. gobio are written without italics.

Response: they are now italicized.

-In order to support the conclusions and go deeper in the evolutionary discussion, I suggest including a phylogenomic approach using a subset of genes of the BUSCO catalog of orthologs for a general tree, and another one more specific for the conserved circadian network genes.

Response: We have now included the suggested phylogenomic/phylogenetic analyses. The results are shown in the new Fig.2 – species phylogeny, and the new Fig. 4 – phylogenetic trees for the four circadian gene families of the core regulatory loop - arntl/BMAL, clock, cry and per, for the 11 notothenioids (representing 7 of the 8 notothenioid families) used in this study.  Two new sections  - 3.3. Phylogenomic analyses affirm E. maclovinus as immediate sister to cryonotothenioids, and 3.5. Phylogenetic analyses of core regulatory loop circadian genes, have been added to describe these results.  For the species tree, instead of using a subset of BUSCO genes, 2,918 single ortholog genes that we were able to identify across all species were used. As expected, the resulting tree confirms the long recognized immediate sister relationship of E. maclovinus to the Antarctic cryonotothenioid clade.  The circadian gene family tree results were discussed with regard to (i) the accuracy and errors in automated curation and annotation, as well as in the context of each circadian paralog’s potential functional role as inferred by the phylogenetic relationship to human orthologs whose functions are well studied.